# Synthesis and Bioactivity of Thiazolethioacetamides as Potential Metallo-β-Lactamase Inhibitors

**DOI:** 10.3390/antibiotics9030099

**Published:** 2020-02-26

**Authors:** Yi-Lin Zhang, Yong Yan, Xue-Jun Wang, Ke-Wu Yang

**Affiliations:** 1College of Biology Pharmacy and Food Engineering, Shangluo University, Shangluo 726000, China; yananan@yeah.net (Y.Y.);; 2Key Laboratory of Synthetic and Natural Functional Molecule Chemistry of Ministry of Education, College of Chemistry and Materials Science, Northwest University, Xi’an 710127, China; kwyang@nwu.edu.cn

**Keywords:** thiazolethioacetamides, metallo-β-lactamase, inhibitor

## Abstract

Metallo-β-lactamase (MβLs) mediated antibiotic resistance seriously threatens the treatment of bacterial diseases. Recently, we found that thioacetamides can be a potential MβL inhibitor skeleton. In order to improve the information of the skeleton, twelve new thiazolethioacetamides were designed by modifying the aromatic substituent. Biological activity assays identify the thiazolethioacetamides can inhibit ImiS with IC_50_ values of 0.17 to 0.70 μM. For two of them, the IC_50_ values against VIM-2 were 2.2 and 19.2 μM, which is lower than in our previous report. Eight of the thiazolethioacetamides are able to restore antibacterial activity of cefazolin against *E.coli*-ImiS by 2–4 fold. An analysis of the structure–activity relation and molecule docking show that the style and position of electron withdrawing groups in aromatic substituents play a crucial role in the inhibitory activity of thiazolethioacetamides. These results indicate that thiazolethioacetamides can serve as a potential skeleton of MβL inhibitors.

## 1. Introduction

Antibacterial resistance has now reached alarming levels worldwide. As classical antibiotics, β-Lactam antibiotics account for over 65% in clinical settings because of their effective antibacterial activity and selectivity [1]. Unfortunately, like other antibacterial drugs, the overuse and misuse of β-lactam antibiotics accelerates the emergence of drug-resistant strains [2]. The main mechanisms of β-lactam antibiotic resistance are due to the production of β-lactamases by bacteria, the resistant strains of which can rapidly spread on a global scale [3]. The β-lactamases can hydrolyze the β-lactam ring, which leads to a loss of antibacterial activity in antibiotics including penicillin, cephalosporins and carbapenems [4].

Based on amino acid sequence homologies, β-lactamases have been categorized into serine β-lactamases (SβLs) and metallo-β-lactamases (MβLs) [5]. SβLs use serine as an active site, existing in successful clinical inhibitors such as sulbactam, tazobactam and clavulanic acid [6]. MβLs can be further categorized into subclasses B1-B3, which utilize either one or two equivalents of Zn(II) as the active site [7]. MβLs currently threaten almost all β-lactamase antibiotics in treating bacterial infection. The rapid diffusion of NDM-1-producing *K.pneumoniae* isolates is a striking example [8]. Because of variations in MβL structures, clinical MβL inhibitors are still lacking. Over the past several years, the literature has provided various approaches for overcoming antibiotic resistance and a number of MβL inhibitors have been designed including sulfonamides, dicarboxylate, β-lactams, cyclic boronates and multivalent chelators [1,9,10,11]. Sulfur-containing compounds occupy an important position in the design of MβL inhibitors because the sulfur atom can reduce the MβLs activity by binding to the zinc ions which are enzyme active center and replacing the bridging water molecules [12,13].

Recently, our group has reported that thioacetamide derivatives exhibit biological activity which may inhibit MβLs [14,15,16,17]. In addition, some of the thioacetamides showed broad-spectrum inhibitory activity against all three subclasses of MβLs. In order to develop the structure–activity relationships, twelve new thiazolethioacetamides were synthesized and characterized. The inhibitory activity was evaluated against MβLs VIM-2, ImiS, and L1, which are representatives of the B1, B2 and B3 subclasses of MβLs, respectively. The ability of these thiazolethioacetamides to defend against the resistant bacterial strain was evaluated by a minimum inhibitory concentrations (MICs) assay. Furthermore, molecular docking was employed when studying the possible interactions between the inhibitors and the corresponding MβLs.

## 2. Results

To acquire effective MβL inhibitors, twelve diaryl-substituted thiazolethioacetamides were synthesized as shown in the Appendix A and characterized by NMR and MS. The yields of the compounds ranged from 56.9% to 87.4% and the structures of these compounds are shown in Figure 1.

To test the inhibitory activity of compounds 1–12 against MβLs, three representative MβLs, VIM-2 (B1), ImiS (B2), and L1 (B3), were chosen for evaluation. The IC_50_ values of the compounds against MβLs with cefazolin as the substrate are listed in Table 1. The inhibition studies indicated that the thiazolethioacetamides had specific inhibitory activity against ImiS and VIM-2, though none of them showed any activity against L1 until the inhibitor concentration reached 1 mM.

It can be observed that compounds 1–12 exhibited an ability to inhibit ImiS with an IC_50_ value range of 0.17–0.70 μM, while compounds 8 and 12 also showed inhibitory activity against VIM-2 with an IC_50_ value 2.2 and 19.2 μM. When the substituent was at the p-position of the benzene ring, compound 8 with trifluoromethoxy had a lower IC_50_ value of 0.31 μM for ImiS than compound 1 containing chlorine (0.58 μM) and compound 9 containing fluorine (0.46 μM). The number and position of chlorine atoms had little effect on the inhibitory activity of compounds 1–4 which had almost the same IC_50_ value. However, with compounds 9–12, fluorine in the p-position improved the ability of the compounds to inhibit ImiS. In addition, compounds 5–7 with chlorine and nitro inhibited ImiS with a better IC_50_ value range of 0.17–0.42 μM, whilst compound 5 gave the lowest IC_50_ value of 0.17 μM among all the compounds. The nitro group may bind selectively with the active site of the enzyme.

The capacity of thiazolethioacetamides to restore the antibacterial activity of cefazolin against *E.coli* BL21 (DE3) cells expressing ImiS and VIM-2 was investigated by determining the minimum inhibitory concentrations (MIC). No compounds had synergistic bacteriostatic effect on *E.coli* and *E.coli*-VIM-2 with cefazolin, and the results to inhibit *E.coli*-ImiS are shown in Table 2. Compounds 5–12 resulted in a 2–4 fold reduction of MIC value for *E.coli*-ImiS in vivo. Inhibitors 1–4 did not change the MIC value relative to the blank control.

In order to explore how the inhibitors bind to MβLs, compounds 8 and 12 were docked into the active pocket of VIM-2 (PDB code 4NQ2), whilst 5 and 8 were docked into CphA (PDB code 2QDS). CphA is an alternative of ImiS which has not been crystallized, because they share a 96% similar sequence. Low-energy conformations (the top ranked conformations) are shown in Figure 2, with binding energies of −6.97, −6.59, −12.64 and −8.14 kcal/mol for the VIM-2/8, VIM-2/12, CphA/5 and CphA/8 complexes, respectively. The molecule docking result reveals the same trend in respect of the IC_50_ values.

According to the bonding mode of the complexes, the docking binding energy of the CphA/inhibitors (ImiS/inhibitors) is significantly lower. This is most likely due to a second Zn(II) ion in VIM-2 resulting in a smaller activity pocket. The bonding energy of VIM-2/8 is lower than that of VIM-2/12, which is probably because the different interactions with Zn(II). In the VIM-2/8 complex, the phenolic hydroxyl group forms hydrogen bonds with Zn(II) and Asn210, with a distance of 3.04 and 2.25 Å, respectively. Moreover, the nitrogen in the thiazole ring and the second Zn(II) interacts with Asp118, whilst the nitrogen in the amido linkage and the oxygen in the trifluoromethoxy group interact with Tyr67 and Arg205, respectively. However, compound 12 exerts inhibitory action only by interacting with oxygen in the amido linkage to form hydrogen bonds with the two Zn(II), with an average distance of 2.29 Å. In complex CphA/inhibitors, the phenolic hydroxyl group of CphA/8 interacts with Zn(II) and Asp120 with a mean distance of 2.94 Å, whilst the oxygen in the amido group forms hydrogen bonds with Asp264. In complex CphA/**5**, the phenolic hydroxyl group interacts with Asp264 and Gln68. The oxygen in the nitro group forms hydrogen bonds with Zn(II), with a distance of 1.88 Å, and His263, with a distance of 2.70 Å, whilst oxygen in the amido group forms bonds with Lys224. The variety of bonds formed increases the affinity with MβLs, which results in the lowest binding energy of −12.64 kcal/mol.

## 3. Discussion

The twelve thiazolethioacetamides exhibit different abilities to inhibit ImiS and VIM-2 with the IC_50_ value range of 0.17–19.2 μM. A possible reason for this is that the kind and position of the substituents in the benzene ring greatly affected the affinity between the inhibitor molecule and MβLs. In line with our previous research [14,15,16,17], electron-absorbing groups have a greater ability to improve the affinity of inhibitors and MβLs. The nitro group has been proven to be an effective zinc ligand in the complex of carboxypeptidase A/aromatic nitropropionic acid [18]. In a molecule docking assay, the nitro group of complex CphA/5 can interact with Zn(II) at short distances. This could be the reason why compounds 5–7 are more effective inhibitors against ImiS. Another factor to consider is the position of these compounds, which depends on the kinds of substituents. Through the analysis of the conformations, the o-position nitro of the benzene ring and the amide can make up a stable virtual six-membered ring which improves the biological activity of compounds 5 and 6. In addition, the biological activity of triazolethioacetamides is better than that of thiazolethioacetamides, but thiazolethioacetamides have a greater potential to inhibit VIM-2. This information is valuable for the further development of MβL inhibitors.

## 4. Materials and Methods

General chemicals were purchased from TCI (Tokyo Chemical Industry, Tokyo, Japan) and were used without further purification. All antibiotics used were purchased from Sigma-Aldrich (Burlington, MO, USA). ^1^H NMR and ^13^C NMR spectra were recorded with a Bruker DRX 600 MHz spectrometer (Bruker Daltonics Inc., Billerica, MA, USA) and a Bruker MicrOTOF-Q II mass spectrometer was used to detect the HRMS data.

### 4.1. Synthesis and Characterization

The synthetic route and spectrum information of the twelve thiazolethioacetamides are shown in the Appendix A. Briefly, N-substituted-2-chloroacetamides and 2-(5-mercapto-1,3,4-thiadiazol-2-yl)phenol were as previously reported [17]. A solution of 2-(5-mercapto-1,3,4-thiadiazol-2-yl)phenol and K_2_CO_3_ (3.6 mmol) dissolved in H_2_O (15 mL) was stirred for 30 min. After N-substituted-2 -chloroacetamides (3 mmol) was added, the reaction mixture was heated to reflux for 6 h. The reaction mixture was cooled to RT and neutralized with 5 M HCl to a pH of approximately 7.0. The resulting white solid was collected by filtration, washed with H_2_O repeatedly (3 × 80 mL) and dried in vacuo to obtain compounds 1–12.

### 4.2. Determination of IC_50_ Values

The enzyme inhibition activity assay was carried out on an Agilent-8453 UV-visible spectrometer. Control experiments verified that the 0.1% DMSO exhibited no inhibitory activity against the MβLs, therefore the final concentrations of DMSO were 0.1% to dissolve thiazolethioacetamides 1–12 in the inhibition experiments and diluted with a 30 mM Tris solution. Cefazolin (Sigma-Aldrich, Burlington, MO, USA), with a concentration of 20 to 140 μM, was taken as the substrate. The enzyme and inhibitor were premixed with the corresponding MβL and the thiazolethioacetamides for 30 min. The IC_50_ values were measured at 25 °C with the inhibitor concentrations of 0 and 1000 μM, and analyzed through IC_50_ calculation tool (AAT Bioquest Inc., sunnyvale, CA, USA).

### 4.3. Determination of MIC Values

According to the Clinical and Laboratory Standards Institute (CLSI) macrodilution (tube) broth method [19], MIC values were determined by using a Thermo Scientific Microplate Reader Multiskan FC. An amount of 1280 µg/mL antibiotics was prepared in ddH_2_O and the compounds 1–12 were dissolved in DMSO to 320 µg/mL solution. MH solution (900 µL) and 100 µL inhibitor solution were added to 2–11 holes of 96-well plates, and an additional 400 µL MH solution, 200 µL inhibitor solution and antibiotic solution (400 µL) were added to the first hole. An amount of 100 µL mix solution was transferred to successive holes. The 100 µL bacterial solution in MH medium was added to each hole. The final concentration of bacterial strains was 5 × 10^5^ CFU per mL. The 96-well plates were incubated at 37 °C for 12 h, and the results were the same as those in the prescribed 16 to 20 h.4.4 Docking Calculations.

Thiazolethioacetamides 5, 8 and 12 were selected as ligand to proceed Docking studies with CphA and VIM-2 by AutoDock 4.2 [20]. The crystal structures of protein were downloaded from the protein database (https://www.rcsb.org) [20]. CphA protein was centered at the location of Zn(II), while VIM-2 was centered at the middle of two Zn(II). The size of grid box was set to 80 × 80 × 80, and the grid points space was at 0.375 Å. During the calculation, the mutation rate and the crossover probability were 0.02 and 0.8, respectively. Meanwhile, other parameters were kept unchanged at the default values. Through the Lamarckian genetic algorithm, each complex produces fifty docking configurations. The lowest-energy (highest ranked) configurations will be further studied.

## 5. Conclusions

In summary, twelve new thiazolethioacetamides were designed and synthesized. Biological activity assays indicate that all the compounds exhibited IC_50_ values ranging from 0.17 to 0.70 μM against MβL ImiS, and two thiazolethioacetamides were able to inhibit VIM-2 with IC_50_ values of 2.2 and 19.2 μM. Eight thiazolethioacetamides could improve the bactericidal capacity of cefazolin to prevent *E.coli*-ImiS by 2–4 fold. Through structure–activity relation and molecule docking, it was revealed that the electron-withdrawing group and position of the substituents are important factors to consider when determining the inhibitory capacity of thiazolethioacetamides. The information in this work is valuable for the further development of MβL inhibitors.

## Figures and Tables

**Figure 1 antibiotics-09-00099-f001:**
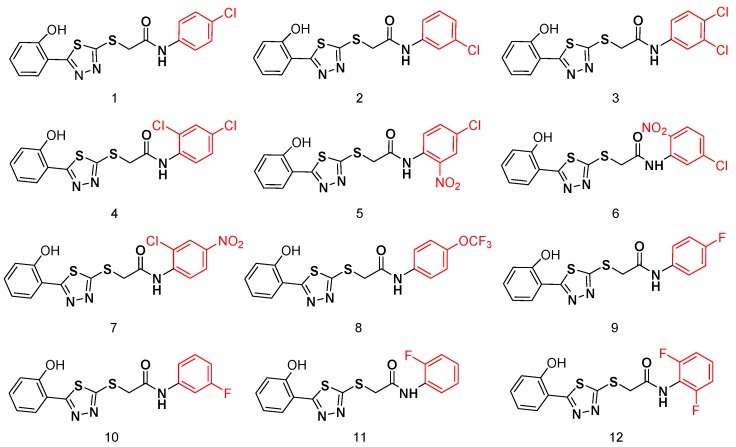
Structures of the synthesized thiazolethioacetamides.

**Figure 2 antibiotics-09-00099-f002:**
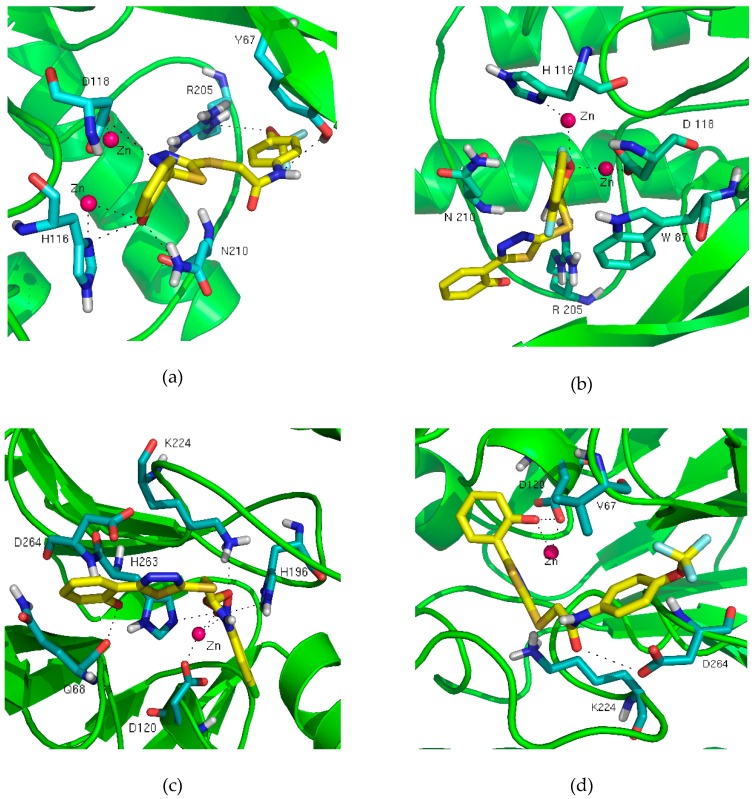
Low energy conformations of compounds 8 (**a**) and 12 (**b**) docked into the active site of VIM-2 (PDB code 4NQ2), 5 (**c**) and 8 (**d**) docked into the active site of CphA (PDB code 2QDS).

**Table 1 antibiotics-09-00099-t001:** IC_50_ values of thiazolethioacetamides against MβLs ImiS and VIM-2.

Compds	IC_50_(μM)		Compds	IC_50_(μM)	
	ImiS	VIM-2		ImiS	VIM-2
1	0.58	-	7	0.42	-
2	0.53	-	8	0.31	2.2
3	0.69	-	9	0.46	-
4	0.58	-	10	0.61	-
5	0.17	-	11	0.68	-
6	0.36	-	12	0.70	19.2

**Table 2 antibiotics-09-00099-t002:** Minimum inhibitory concentrations (MIC)(μg/mL) value of cefazolin against *E. coli*-ImiS in the presence of thiazolethioacetamides.

Compds	*E.coli*-ImiS	Compds	*E.coli*-ImiS	Compds	*E.coli*-ImiS
Blank	20	5	10	10	5
1	20	6	10	11	10
2	20	7	10	12	10
3	20	8	5		
4	20	9	5

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
