# Peer review of "Synthesis and Bioactivity of Thiazolethioacetamides as Potential Metallo-β-Lactamase Inhibitors"

_antibiotics, 2020, doi:10.3390/antibiotics9030099_

Round 1
Reviewer 1 Report
I think that the work is well writen and well designed. I just have some doubts that I marked in the attached document.

Author Response
Response Letter
Dear Reviewer and Editor,
We have studied the valuable comments from you carefully, and tried our best to revise the manuscript. The point to point responds to the reviewer’s comments are listed as following:
Comment 1: I think that in this figure, the authors should present also the Yield of the sinthesys.
Response: We showed the yield of all the sinthesys in the Supporting Information, and we also supplement it appropriately in Line 57-58 of the manuscript as following:
“Toward to acquire effective MβLs inhibitors, twelve diaryl-substituted thiazolethioacetamides were synthesized as shown in Supporting Information and characterized by NMR and MS. The yields of the compounds were ranged from 56.9% to 87.4%, and the structures of these compounds were shown in Figure 1.”
Comment 2: Where the bacteria come from? Is an ATCC?
Response: We described the source of the bacteria in Acknowledgments “…… All the enzymes and bacteria used in the study were donated from Professor Kewu Yang.” The E.coli (ATCC25922) and the drug resistant plasmid was presented to professor Yang as gifts from professor Crowder.
Comment 3: Mark the sentence “Based on the low concentration of DMSO” in yellow.
Response: The sentence was devised in Chapter 4.2 as the following:
“The enzyme inhibition activity assay was carried out on Agilent-8453 UV-visible spectrometer. Control experiments verified that the 0.1% DMSO had no inhibitory activity against the MβLs, therefore the final concentrations of DMSO were 0.1% to dissolve thiazolethioacetamides 1-12 in inhibition experiments and diluted with a 30 mM Tris solution.”
Comment 4: Mark the “macrodilution” in the sentence “According to the Clinical and Laboratory Standards Institute (CLSI) macrodilution (tube) broth method”.
Response: The “macrodilution” was introduced from the references “J Nat Med, 2015, 69, 241-248” and “Ind Crop Prod, 2013, 51, 93-99.”
We look forward to hearing from you regarding our submission. We would be glad to respond to any further questions and comments that you may have.
Best wishes,
Yilin Zhang

Reviewer 2 Report
In the era of increasing drug resistance of microorganisms, it is extremely important to look for new sources of substances with antimicrobial activity or improvement of existing antibiotics. The synthesis of the inhibitors of enzymes that break down antibiotics is an important challenge for modern science. Enzyme inhibitors, such as those presented in the paper, may constitute an effective strategy for treating infections caused by the strains resistant to beta-lactam antibiotics. In this regard, the topic of the reviewed work is important for the development of the field.
Detailed comments:
Chapter 4.3
Please indicate the final bacterial density in CFU/ml that was achieved in each well of the 96-well plate.
Please enter the final DMSO concentration in the wells. Was growth control performed with the addition of DMSO?
Why did the authors incubate the bacteria for 12 hours with tested antibiotic and inhibitors? The CLSI standard indicates incubation from 16 to 20 hours?
Author Response
Response Letter
Dear Reviewer and Editor,
We have studied the valuable comments from you carefully, and tried our best to revise the manuscript. The point to point responds to the reviewer’s comments are listed as following:
Comment 1: Chapter 4.3 Please indicate the final bacterial density in CFU/ml that was achieved in each well of the 96-well plate.
Response: The final bacterial density has been added in Chapter 4.3. The content is “……The 100 µL bacterial solution in MH medium was added to each hole. The final concentration of bacterial strains was 5×105 CFU per mL.”
Comment 2: Please enter the final DMSO concentration in the wells. Was growth control performed with the addition of DMSO?
Response: The control group and the experimental group contained the same amount of DMSO, and both were subjected to inhibition tests under the same conditions. The final DMSO concentration in the wells in Chapter 4.2. The content is as following:
“The enzyme inhibition activity assay was carried out on Agilent-8453 UV-visible spectrometer. Control experiments verified that the 0.1% DMSO had no inhibitory activity against the MβLs, therefore the final concentrations of DMSO were 0.1% to dissolve thiazolethioacetamides 1-12 in inhibition experiments and diluted with a 30 mM Tris solution.”
Comment 3: Why did the authors incubate the bacteria for 12 hours with tested antibiotic and inhibitors? The CLSI standard indicates incubation from 16 to 20 hours?
Response: The MIC results were no different by monitoring the growth of bacteria at 12, 14, 16 and 20 h, respectively. Therefore, the 12 hours as the shortest time was selected in the manuscript. We also carried on the supplement as following:
“The 96-well plates were incubated at 37℃ for 12 h, and the results were the same as those in the prescribed 16 to 20 h.”
We look forward to hearing from you regarding our submission. We would be glad to respond to any further questions and comments that you may have.
Best wishes,
Yilin Zhang
